# A*+BFHS: A Hybrid Heuristic Search Algorithm

## Zhaoxing Bu, Richard E. Korf

Computer Science Department
University of California, Los Angeles
Los Angeles, CA 90095
{zbu, korf}@cs.ucla.edu

## Abstract

We present a new algorithm A*+BFHS for solving problems where A* and IDA* fail due to memory limitations and/or the existence of many short cycles. A*+BFHS is based on A* and breadth-first heuristic search (BFHS). A*+BFHS combines advantages from both algorithms, namely A*'s node ordering, BFHS's memory savings, and both algorithms' duplicate detection. On easy problems, A*+BFHS behaves the same as A*. On hard problems, it is slower than A* but saves a large amount of memory. Compared to BFIDA*, A*+BFHS reduces the search time and/or memory requirement by several times on a variety of planning domains.

## Introduction and Overview

A* (Hart, Nilsson, and Raphael 1968) is a classic heuristic search algorithm that is used by many state-of-the-art optimal track planners (Katz et al. 2018; Franco et al. 2017, 2018; Martinez et al. 2018). One advantage of A* is duplicate detection. A* uses a Closed list and an Open list to prune duplicate nodes. A state is a unique configuration of the problem while a node is a data structure that represents a state reached by a particular path. Duplicate nodes represent the same state arrived at via different paths.

The second advantage of A* is node ordering. A* always picks an Open node whose $f$-value is minimum among all Open nodes to expand next, which guarantees an optimal solution returned by A* when using an admissible heuristic. When using a consistent heuristic, A* expands all nodes whose $f$-value is less than the optimal solution cost ($C^*$). However, tie-breaking among nodes of equal $f$-value significantly affects the set of expanded nodes whose $f$-value equals $C^*$. It is common practice to choose an Open node whose $h$-value is minimum among all Open nodes with the same $f$-value, as this strategy usually leads to fewer nodes expanded. A survey of tie-breaking strategies in A* can be found in (Asai and Fukunaga 2016).

A*'s main drawback is its exponential space requirement as it stores in memory all nodes generated during the search. For example, A* can fill up 8 GB of memory in a few minutes on common heuristic search and planning domains. To solve hard problems where A* fails due to memory limitations, researchers have proposed various algorithms, usually by forgoing A*'s duplicate detection or node ordering. For example, Iterative-Deepening-A* (IDA*, Korf 1985) only

has a linear memory requirement, at the price of no duplicate detection and a depth-first order within each search bound. However, IDA* may generate too many duplicate nodes on domains containing lots of short cycles, such as Towers of Hanoi and many planning domains, limiting its application.

This paper introduces a new algorithm for solving hard problems with many short cycles, where IDA* is not effective. First, we review previously developed algorithms. Second, we present our algorithm A*+BFHS, which is based on A* and Breadth-First Heuristic Search (Zhou and Hansen 2004). Third, we present experimental results on 32 hard instances from 18 International Planning Competition (IPC) domains. On those problems, A*+BFHS is slower than A* but requires significantly less memory. Compared to BFIDA*, which is an algorithm that requires less memory than A*, A*+BFHS reduces the search time and/or memory requirement by several times, and sometimes by an order of magnitude, on a variety of domains.

## Previous Work

IDA* with a transposition table (IDA*+TT, Sen and Bagchi 1989; Reinefeld and Marsland 1994) uses a transposition table to detect duplicate nodes. However, IDA*+TT is outperformed by other algorithms on both heuristic search (Bu and Korf 2019) and planning domains (Zhou and Hansen 2004).

A*+IDA* (Bu and Korf 2019) combines A* and IDA*, and is the state-of-the-art algorithm on the 24-Puzzle. It first runs A* until memory is almost full, then runs IDA* below each frontier node without duplicate detection. By sorting the frontier nodes with the same $f$-value in increasing order of $h$-values, A*+IDA* can significantly reduce the number of nodes generated in its last iteration. Compared to IDA*, we reported a reduction by a factor of 400 in the total number of nodes generated in the last iteration on all 50 24-Puzzle test cases in (Korf and Felner 2002). Similar to IDA*, A*+IDA* does not work well on domains with many short cycles, however, as in many planning domains.

Frontier search (Korf et al. 2005) is a family of heuristic search algorithms that work well on domains with many short cycles. Rather than storing all nodes generated, it stores only nodes that are at or near the search frontier, including all Open nodes and only one or two layers of Closed nodes. As a result, when a goal node is expanded, only the optimal cost is known. To reconstruct the solution path, fron-

tier search keeps a middle layer of Closed nodes in memory. For example, we can save the Closed nodes at depth $h(start)/2$ as the middle layer. Each node generated below this middle layer has a pointer to its ancestor in the middle layer. After discovering the optimal cost, a node in the middle layer that is on an optimal path is identified. Then the same algorithm can be applied recursively to compute the solution path from the start node to the middle node, and from the middle node to the goal node. In general, however, frontier search cannot prune all duplicates in directed graphs (Korf et al. 2005; Zhou and Hansen 2004).

Divide-and-Conquer Frontier-A* (DCFA*, Korf and Zhang 2000) is a best-first frontier search based on A*. To reconstruct the solution path, DCFA* keeps a middle layer of Closed nodes that are roughly halfway along the solution path. DCFA* detects duplicates and maintains A*'s node ordering, but its memory savings compared to A* is limited on domains where the Open list is larger than the Closed list.

Breadth-First Heuristic Search (BFHS, Zhou and Hansen 2004) is a frontier search algorithm for unit-cost domains. BFHS also detects duplicates but uses a breadth-first node ordering instead of A*'s best-first ordering. At first, assume the optimal cost $C^*$ is known in advance. BFHS runs a breadth-first search (BFS) from the start node and prunes every generated node whose $f$-value exceeds $C^*$. To save memory, BFHS only keeps a few layers of nodes in memory. On undirected graphs, if we store the operators used to generate each node, and do not regenerate the parents of a node via the inverses of those operators, frontier search only needs to store two layers of nodes, the currently expanding layer and their child nodes (Korf et al. 2005). On directed graphs, one previous layer besides the above-mentioned two layers is usually stored to detect duplicates (Zhou and Hansen 2004). To reconstruct the solution path, Zhou and Hansen (2004) recommend saving the layer at the 3/4 point of the solution length as the middle layer instead of the layer at the halfway point, which usually requires more memory. As shown in (Zhou and Hansen 2004), on a domain where the Open list of A* is larger than the Closed list, BFHS usually ends up storing fewer nodes than DCFA*.

In general, $C^*$ is not known in advance. Breadth-First Iterative-Deepening-A* (BFIDA*, Zhou and Hansen 2004) overcomes this issue by running multiple iterations of BFHS, each with a different $f$-bound, starting with the heuristic value of the start node. Similar to IDA*, the last iteration of BFIDA* is often significantly larger than previous iterations, so most search time is spent on the last iteration on many domains.

Compared to A*, BFHS and BFIDA* save significant memory but generate more nodes. The main drawback of BFHS and BFIDA* is that their node ordering is almost the worst among different node ordering schemes. BFHS and BFIDA*'s breadth-first ordering means they have to expand all nodes stored at one depth before expanding any nodes in the next depth. As a result, they have to expand almost all nodes whose $f$-value equals $C^*$, excepting only some nodes at the same depth as the goal node, while A* may only expand a small fraction of such nodes due to its node ordering.

Forward Perimeter Search (FPS, Schütt, Döbbelin, and

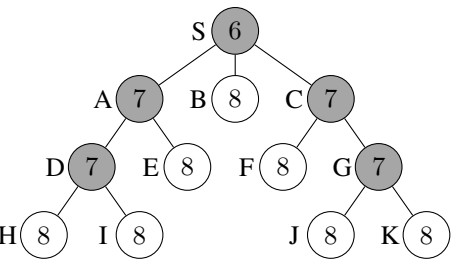

Figure 1: An example of A*+BFHS's search frontier. Numbers are $f$-values. Closed nodes are gray.

Reinefeld 2013) builds a perimeter around the start node via BFS, then runs BFIDA* below each perimeter node. The authors only test FPS on the 24-Puzzle and 17-Pancake problem, and did not report any running times.

## A*+BFHS

### Algorithm Description

We propose a hybrid algorithm we call A*+BFHS to solve hard problems with many short cycles. A*+BFHS first runs A* until a storage threshold is reached, then runs a series of BFHS iterations on sets of frontier nodes, which are the Open nodes at the end of the A* phase.

The BFHS phase can be viewed as a doubly nested loop. Each iteration of the outer loop, which we define as an iteration of the BFHS phase, corresponds to a different cost bound for BFHS. The first cost bound is set to the smallest $f$-value among all frontier nodes. In each iteration of the BFHS phase, we first partition the frontier nodes whose $f$-value equals the cost bound into different sets according to their depths. Then the inner loop makes one call to BFHS on each set of frontier nodes, in decreasing order of their depths. This is done by initializing the BFS queue of each call to BFHS with all the nodes in the set. This inner loop continues until a solution is found or all calls to BFHS with the current bound fail to find a solution. After each call to BFHS on a set of frontier nodes, we increase the $f$-value of all nodes in the set to the minimum $f$-value of the nodes generated but not expanded in the previous call to BFHS.

Figure 1 presents an example of the Open and Closed nodes at the end of the A* phase. Node S is the start node. All edge costs are 1 and the number in each node is its $f$-value. Closed nodes are gray. The Open nodes B, E, F, H, I, J, K are the frontier nodes for the BFHS phase. A*+BFHS first makes a call to BFHS with a cost bound of 8 on all frontier nodes at depth 3, namely nodes H, I, J, K. If no solution is found, A*+BFHS updates the $f$-values of all these nodes to the minimum $f$-value of the nodes generated but not expanded in that call to BFHS. A*+BFHS then makes a second call to BFHS with bound 8, starting with all frontier nodes at depth 2, namely nodes E and F. If no solution is found, A*+BFHS updates the $f$-values of these nodes, then makes a third call to BFHS with bound 8, starting with the frontier node B at depth 1. Suppose that no solution is found with bound 8, the updated $f$-values for nodes E, F, H, I, J, K are 9, and the updated $f$-value for node B is 10. A*+BFHS

then starts a new iteration of BFHS with a cost bound of 9, making two calls to BFHS on nodes at depth 3 and 2 respectively. If the solution is found in the first call to BFHS with bound 9, BFHS will not be called again on nodes E and F.

A*+BFHS is complete and admissible when using an admissible heuristic. A*+BFHS potentially makes calls to BFHS on all frontier nodes. When an optimal solution exists, one node on this optimal path will serve as one of the start nodes for one of the calls to BFHS. Such a node is guaranteed to exist by A*'s completeness and admissibility. Then when the cost bound for the calls to BFHS equals $C^*$, the optimal solution will be found, guaranteed by BFHS's completeness and admissibility.

A state can be regenerated in separate calls to BFHS in the same iteration. To reduce such duplicates, we can decrease the number of calls to BFHS in each iteration by making each call to BFHS on a combined set of frontier nodes at adjacent depths. For the example in Figure 1, we can make one call to BFHS on the frontier nodes at depths 2 and 3 together instead of two separate calls to BFHS, by putting the frontier nodes at depth 3 after the frontier nodes at depth 2 in the initial BFS queue.

In practice, we can specify a maximum number of calls to BFHS per iteration. Then in each iteration, we divide the number of depths of the frontier nodes by the number of calls to BFHS to get the number of depths for each call to BFHS. For example, if the depths of the frontier nodes range from 7 to 12 and we are limited to three calls to BFHS per iteration, each call to BFHS will start with frontier nodes at two depths. We used this strategy in our experiments.

For each node generated in the BFHS phase, we check if it was generated in the A* phase. If so, we immediately prune the node if its current $g$-value in the BFHS phase is greater than or equal to its stored $g$-value in the A* phase.

The primary purpose of the A* phase is to build a frontier set, so that A*+BFHS can terminate early in its last iteration. In the A* phase we have to reserve some memory for the BFHS phase. In our experiments, we first generated pattern databases or the merge-and-shrink heuristic, then allocated 1/10 of the remaining memory of 8 GB for the A* phase.

## Comparisons to BFIDA* and FPS

A*+BFHS's BFHS phase also uses the iterative deepening concept of BFIDA*, but there are two key differences. First, in each iteration, BFIDA* always makes one call to BFHS on the start node, while we call BFHS multiple times, each on a different set of frontier nodes. Second, in each iteration, we order the frontier nodes based on their depth, and run BFHS on the deepest frontier nodes first.

These differences lead to one drawback and two advantages. The drawback is that A*+BFHS may generate more nodes than BFIDA*, as the same state can be regenerated in separate calls to BFHS in the same iteration.

The first advantage is that A*+BFHS may terminate early in its last iteration. If A*+BFHS generates a goal node in the last iteration below a relatively deep frontier node, no frontier nodes above that depth will be expanded. Therefore, A*+BFHS may generate only a small number of nodes in its last iteration. In contrast, BFIDA* has to expand almost all

nodes whose $f$-value is less than or equal to $C^*$ in its last iteration. As a result, A*+BFHS can be faster than BFIDA*.

The second advantage is that A*+BFHS's memory usage, which is the maximum number of nodes stored during the entire search, may be smaller than that of BFIDA* for two reasons. First, the partition of frontier nodes and separate calls to BFHS within the same iteration can reduce the maximum number of nodes stored in the BFHS phase. Second, BFIDA* stores the most nodes in its last iteration while A*+BFHS may store only a small number of nodes in the last iteration due to early termination. Thus, A*+BFHS may store the most nodes in the penultimate iteration instead.

FPS looks similar to A*+BFHS, but there are several fundamental differences. First, FPS builds the perimeter using a breadth-first approach while A*+BFHS builds the frontier via a best-first approach. FPS can also dynamically extend the perimeter but this approach does not always speed up the search (Schütt, Döbbelin, and Reinefeld 2013). Second, in each iteration of FPS's BFIDA* phase, FPS makes one call to BFHS on each perimeter node. In contrast, in A*+BFHS each call to BFHS is on a set of frontier nodes. Third, FPS sorts the perimeter nodes at the same $f$-value using a max-tree-first or longest-path-first policy, while A*+BFHS sorts the frontier nodes at the same $f$-value in decreasing order of their depth. Fourth, FPS needs two separate searches for solution reconstruction while A*+BFHS only needs one.

## Solution Reconstruction

Each node generated in A*+BFHS's BFHS phase has a pointer to its ancestral frontier node. When a goal node is generated, the solution path from the start node to the ancestral frontier node is stored in the A* phase and only one more search is needed to reconstruct the solution path from the ancestral frontier node to the goal node. This subproblem is much easier than the original problem and we can use the same heuristic function as for the original problem. Therefore, we just use A* to solve this subproblem. In addition, since we know the optimal cost of this subproblem, we can prune any node whose $f$-value exceeds this cost.

In BFIDA*, we have to solve two subproblems to recover the solution path from the start node to the middle node and from the middle node to the goal node. Zhou and Hansen (2004) called BFHS recursively to solve these two subproblems. However, pattern database heuristics (PDB, Culberson and Schaeffer 1998) only store heuristic values to the goal state, and not between arbitrary pairs of states, which complicates finding a path to a middle node. Similar to A*+BFHS, we use A* to solve the second subproblem. For the first subproblem, we use A* to compute the path from the start node to the middle node using the same heuristic function as for the original problem, which measures the distance to the goal node, not the middle node. To save memory, we prune any node whose $g$-value is greater than or equal to the depth of the middle node, and any node whose $f$-value exceeds the optimal cost of the original problem. Since a deeper middle layer leads to more nodes stored in this approach, we saved the layer at the 1/4 point of the solution length as the middle layer instead of the 3/4 point. In this way, we do not need to build a new heuristic function

for the middle node. In our experiments, the search time for solution reconstruction in BFIDA* is usually less than 1% of the total search time.

## Experimental Results and Analysis

We implemented BFIDA* and A*+BFHS in the planner Fast Downward 20.06 (Helmert 2006), using the existing code for node expansion and heuristic value lookups. A*+BFHS's A* phase reused the existing A* code. A* stores all nodes in one hash map. We used the same hash map implementation with the following difference. In each call to BFHS in both BFIDA* and A*+BFHS, we saved three layers of nodes for duplicate detection and we created one hash map for each layer of nodes. We did this because storing all nodes in one hash map in BFHS involves a lot of overhead, and is more complicated. Schütt, Döbbelin, and Reinefeld (2013) did not test FPS on planning domains and we do not know the optimal perimeter radius and sorting strategy for each domain, so we did not implement FPS in Fast Downward.

We solved about 550 problem instances from 32 unit-cost domains. We present the results of A*, BFIDA*, and A*+BFHS on the 32 hardest instances. All remaining instances were easily solved by A*. We tested two A*+BFHS versions. A*+BFHS ($\infty$) starts each call to BFHS on frontier nodes at one depth. A*+BFHS (4) makes each call to BFHS on frontier nodes at multiple depths with at most four calls to BFHS in each iteration. All tests were run on a 3.33 GHz Intel Xeon X5680 CPU with 236 GB of RAM. We used the landmark-cut heuristic (LM-cut, Helmert and Domshlak 2009) for the satellite domain, the merge-and-shrink heuristic (M&S) with the recommended configuration (Sievers, Wehrle, and Helmert 2014, 2016; Sievers 2018) for the tpp and hiking14 domains, and the iPDB heuristic with the default configuration (Haslum et al. 2007; Sievers, Ortlieb, and Helmert 2012) for all other domains.

We present the results in Tables 1, 2, and 3. Tables 1 and 2 contain the 26 hardest instances solved by A*. Table 3 contains the remaining 6 instances where A* terminated early without finding a solution due to the limitation of the hash map size in Fast Downward 20.06. The instances in Tables 1 and 2 are sorted by the A* running times and the instances in Table 3 are sorted by the BFIDA* running times.

All three tables have the same columns. The first column gives the domain name, the instance ID, the optimal solution cost $C^*$, and the heuristic function used. The second column lists the different algorithms. We ran each algorithm until it found an optimal cost and returned the optimal path. The third column gives the maximum number of nodes stored by each algorithm. For A*, this is the number of nodes stored at the end of the search. For BFIDA*, this is the largest sum of the number of nodes stored in all three layers of the search, plus the nodes stored in the 1/4 layer for solution reconstruction. For A*+BFHS, this is the largest number of nodes stored in the BFHS phase plus the number of nodes stored in the A* phase. An underline means the specific algorithm needed more than 8 GB of memory to solve the problem. The fourth column is the total number of nodes generated, including the nodes generated during solution reconstruction. The fifth column is the number of nodes generated in all

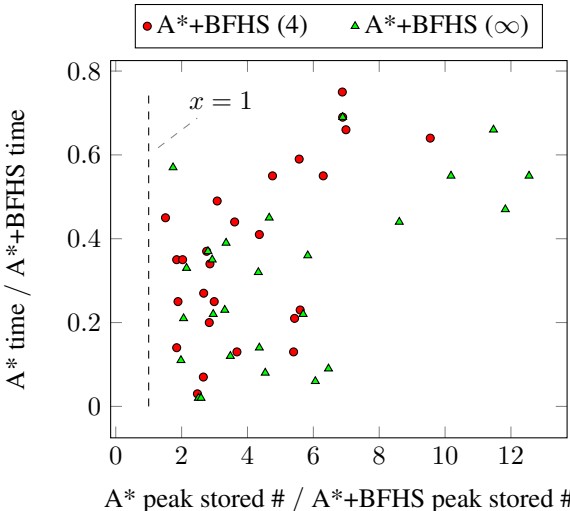

Figure 2: A* vs. A*+BFHS in time and memory.

but the last iteration. For A*, this is the number of nodes generated before expanding an Open node whose $f$-value is $C^*$. For A*+BFHS, this number includes the nodes generated in its A* phase. The sixth column is the number of nodes generated in the last iteration. For A*, this is the number of nodes generated while expanding the Open nodes whose $f$-value equals $C^*$. The last column is the running time in seconds, including the time for solution reconstruction but excluding the time spent on precomputing the heuristic function, which is the same for all algorithms. For each instance, the smallest maximum number of stored nodes and shortest running time are indicated in boldface. For the A* data in Table 3, we report the numbers of nodes and running times just before A* terminated, with a $>$ symbol to indicate such numbers.

We further compare the time and memory between A* and A*+BFHS in Figure 2, and between BFIDA* and A*+BFHS in Figure 3, where the $x$-axis is A*/BFIDA*'s peak stored nodes over A*+BFHS's and the $y$-axis is A*/BFIDA*'s running time over A*+BFHS's. Figure 2 contains the 26 instances solved by A* and Figure 3 contains all 32 instances. The red circles and green triangles correspond to A*+BFHS (4) and A*+BFHS ($\infty$) respectively. The data points above the $y = 1$ line or to the right of the $x = 1$ line represent instances where A*+BFHS outperformed the comparison algorithm in terms of time or memory.

### A*+BFHS vs. A*

A* was the fastest on all problem instances that it solved, but also used the most memory. Among the 32 hardest problem instances we present, A* required more than 8 GB of memory on 22 instances and could not find a solution on 6 of those after running out of the hash map used by Fast Downward 20.06. On some of these instances, A* used 30 GB to 40 GB of memory before it terminated. This means A* cannot solve these 22 instances under the current IPC memory requirement, which is 8 GB. A*+BFHS required several times, sometimes an order of magnitude, less mem-

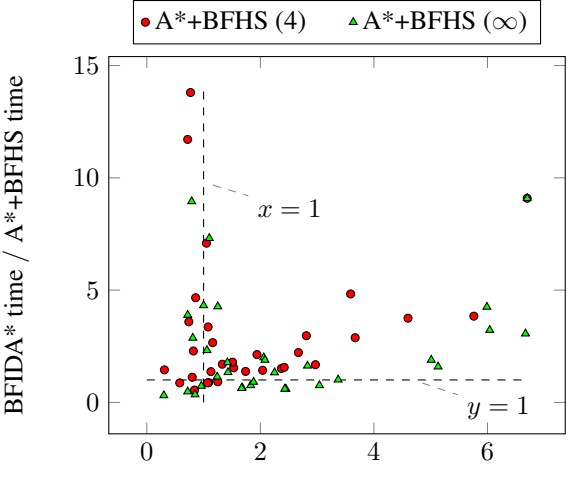

Figure 3: BFIDA* vs. A*+BFHS in time and memory.

ory than A*. As a result, A*+BFHS only used more than 8 GB of memory on one instance. An interesting comparison is the space and time trade-off. For example, on parking14, A*+BFHS increased the running time by less than 100% while saving more than an order of magnitude in memory.

## A*+BFHS vs. BFIDA*

In summary, on easy problems that A*+BFHS can solve in its A* phase, A*+BFHS behaves the same as A*, and is always faster than BFIDA*. We solved around 500 such problems, which are not included here due to space limitations. On the 32 hardest problems we present, A*+BFHS is faster than BFIDA* on 27 instances and at least twice as fast on 16 of those. Furthermore, A*+BFHS requires less memory than BFIDA* on 25 of the 32 instances and saves more than half the memory on 14 of those. In addition, these time and memory reductions exist on both the relatively easy and hard ones of the 32 instances presented, demonstrating that A*+BFHS is in general better than BFIDA* on very hard problems as well as easy problems. In the following paragraphs, we compare A*+BFHS with BFIDA* in four aspects: duplicate detection, node ordering, memory, and running times.

The relative numbers of nodes generated in the previous iterations reflect the power of duplicate detection. Compared to BFIDA*, A*+BFHS (4) generated a similar number of nodes in the previous iterations on most instances. Hiking14 2-3-6 is the only instance where A*+BFHS (4) generated at least twice as many nodes in the previous iterations as BFIDA*. However, A*+BFHS ($\infty$) generated 2 to 7 times as many nodes in the previous iterations as BFIDA* on 11 instances. This contrast shows that, compared to BFIDA*, significantly more duplicate nodes can be generated by making each call to BFHS on frontier nodes at only one depth. However, most of those duplicate nodes can be avoided by making each call to BFHS on frontier nodes at multiple depths.

A*+BFHS can generate fewer duplicate nodes than BFIDA* due to fewer BFHS iterations and making each call

to BFHS on a set of frontier nodes. A*+BFHS reduced the number of nodes in previous iterations by around 50% on freecell 06 and snake18 17, and a factor of 4 on snake18 08. To our surprise, we found that on snake18 08, the number of nodes generated in the penultimate iteration of BFIDA* was twice as many as the sum of the nodes generated in A*+BFHS's A* phase and the penultimate iteration of the BFHS phase. This means a lot of duplicate nodes were generated in BFIDA*. Snake18 generates a directed graph, in which case frontier search cannot detect all duplicate nodes (Korf et al. 2005; Zhou and Hansen 2004).

Compared to BFIDA*, A*+BFHS reduced the number of nodes in the last iteration significantly, and usually by several orders of magnitude, on 28 of the 32 instances. This large reduction proves that when ordering the frontier nodes by deepest-first, A*+BFHS can terminate early in its last iteration. On the three blocks instances and depot 11, A*+BFHS did not terminate early in its last iteration because the ancestral frontier node of the goal had a relatively low $g$-value. In fact, A* generated the most nodes in its last iteration on the three blocks instances, which shows that node ordering is also difficult for A* on those instances. In contrast, A* generated very few nodes in its last iteration on depot 11, suggesting that A*+BFHS may terminate early in its last iteration given more memory for its A* phase.

A*+BFHS's A* phase usually stored from 10 to 20 million nodes, with the exception of the snake18 domain where 40 to 50 million nodes were stored. Comparing the maximum number of stored nodes, A*+BFHS ($\infty$) required less memory than BFIDA* on 25 instances and less than half the memory on 14 of those. For A*+BFHS (4), these two numbers are 23 and 11 respectively. In contrast, termes18 05 is the only instance where the maximum number of stored nodes of A*+BFHS was at least twice that of BFIDA*.

Comparing the two versions of A*+BFHS, A*+BFHS (4) was usually faster, sometimes significantly, due to the reduction in duplicate nodes. Compared to BFIDA*, A*+BFHS (4) was slightly slower on four instances and 80% slower on one instance. On the other 27 instances, A*+BFHS was faster than BFIDA*, and at least twice as fast on 16 of those. The large speedups usually were on the instances where BFIDA* generated the most nodes in its last iteration. The best result was on the logistics00 domain, where an order of magnitude speedup was achieved. This is because BFIDA* performed very poorly on this domain due to its breadth-first node ordering. Comparing A*+BFHS ($\infty$) with BFIDA*, A*+BFHS ($\infty$) was slower on 11 instances and at least twice as slow on three of those, but also at least twice as fast on 12 instances. The main reason for the slower cases is the presence of many duplicate nodes generated in certain domains.

## Calling BFHS on Nodes at Multiple Depths

Comparing the two A*+BFHS versions, each has its pros and cons. A*+BFHS (4) always generated fewer duplicate nodes. Comparing the number of nodes generated in the previous iterations, A*+BFHS ($\infty$) generated at least twice as many nodes on 7 instances. A*+BFHS ($\infty$) generated significantly fewer nodes in the last iteration than A*+BFHS (4) on 22 instances. However, the number of nodes generated in

| Instance | Algorithm | Peak stored | Total nodes | Prev. iterations | Last iteration | Time (s) |
|---|---|---|---|---|---|---|
| *depot* | A* | 70,504,763 | 344,658,749 | 344,639,234 | 19,515 | **233** |
| 14 | BFIDA* | **17,042,841** | 1,390,466,785 | 582,348,193 | 795,336,992 | 1,708 |
| $C^*$=29 | A*+BFHS ($\infty$) | 21,023,657 | 556,674,817 | 540,764,124 | 15,909,899 | 596 |
| iPDB | A*+BFHS (4) | 22,882,537 | 446,204,987 | 432,278,188 | 13,926,005 | 475 |
| *termes18* | A* | 80,012,545 | 211,514,579 | 211,514,568 | 11 | **245** |
| 05 | BFIDA* | **9,370,587** | 3,757,844,868 | 3,413,500,020 | 221,186,298 | 4,796 |
| $C^*$=132 | A*+BFHS ($\infty$) | 30,874,300 | 10,702,979,649 | 10,701,959,808 | 911,786 | 15,415 |
| iPDB | A*+BFHS (4) | 30,076,170 | 2,271,661,960 | 2,270,262,609 | 1,291,296 | 3,319 |
| *freecell* | A* | 53,080,996 | 243,947,771 | 243,244,703 | 703,068 | **250** |
| 06 | BFIDA* | 38,054,162 | 1,220,132,074 | 732,920,409 | 485,268,534 | 1,883 |
| $C^*$=34 | A*+BFHS ($\infty$) | **30,481,377** | 327,209,951 | 312,812,283 | 14,388,579 | 441 |
| iPDB | A*+BFHS (4) | 35,120,076 | 403,465,250 | 302,581,091 | 100,875,070 | 561 |
| *logistics00* | A* | 57,689,357 | 107,083,712 | 106,929,666 | 154,046 | **255** |
| 14-1 | BFIDA* | **15,441,813** | 3,137,204,256 | 106,929,666 | 3,020,315,591 | 10,381 |
| $C^*$=71 | A*+BFHS ($\infty$) | 19,472,255 | 354,438,805 | 354,058,774 | 368,595 | 1,160 |
| iPDB | A*+BFHS (4) | 20,169,648 | 227,903,318 | 110,674,320 | 117,217,562 | 752 |
| *driverlog* | A* | 144,065,288 | 420,609,830 | 420,609,777 | 53 | **344** |
| 12 | BFIDA* | 35,034,406 | 1,718,350,515 | 678,644,177 | 1,030,180,074 | 1,676 |
| $C^*$=35 | A*+BFHS ($\infty$) | **24,712,720** | 1,020,438,794 | 1,020,410,754 | 27,959 | 944 |
| iPDB | A*+BFHS (4) | 30,270,816 | 643,723,984 | 641,790,459 | 1,933,444 | 631 |
| *freecell* | A* | 107,183,015 | 531,379,136 | 531,378,858 | 278 | **522** |
| 07 | BFIDA* | 77,196,602 | 4,152,881,254 | 2,897,339,576 | 1,143,762,584 | 6,416 |
| $C^*$=41 | A*+BFHS ($\infty$) | **54,171,433** | 3,095,608,289 | 2,370,094,738 | 725,267,629 | 4,775 |
| iPDB | A*+BFHS (4) | 58,058,327 | 2,430,947,097 | 1,896,369,611 | 534,331,564 | 3,769 |
| *depot* | A* | 172,447,963 | 764,608,339 | 764,607,971 | 368 | **550** |
| 11 | BFIDA* | **27,192,174** | 3,037,154,042 | 1,260,718,486 | 1,755,157,316 | 3,544 |
| $C^*$=46 | A*+BFHS ($\infty$) | 37,977,775 | 6,268,318,349 | 3,092,746,859 | 3,175,552,575 | 7,314 |
| iPDB | A*+BFHS (4) | 46,923,423 | 3,319,995,622 | 1,262,429,685 | 2,057,547,022 | 4,078 |
| *tpp* | A* | 187,011,066 | 610,996,630 | 610,995,018 | 1,612 | **562** |
| 11 | BFIDA* | 93,759,836 | 4,290,825,940 | 754,905,369 | 3,525,135,895 | 7,214 |
| $C^*$=51 | A*+BFHS ($\infty$) | **30,856,159** | 5,504,314,294 | 5,504,268,064 | 46,111 | 9,550 |
| M&S | A*+BFHS (4) | 33,368,912 | 1,419,143,562 | 1,285,410,734 | 133,732,709 | 2,426 |
| *mystery* 14 | A* | 139,924,686 | 652,569,481 | 650,036,341 | 2,533,140 | **578** |
| $C^*$=11 | BFIDA* | 135,963,227 | 6,213,135,253 | 727,753,687 | 5,430,082,105 | 7,628 |
| iPDB | A*+BFHS ($\infty$/4) | **20,302,860** | 730,971,724 | 676,473,465 | 54,497,630 | 839 |
| *tidybot11* | A* | 69,953,936 | 171,363,621 | 170,286,720 | 1,076,901 | **662** |
| 17 | BFIDA* | 42,080,838 | 776,084,110 | 486,518,217 | 281,131,278 | 3,684 |
| $C^*$=40 | A*+BFHS ($\infty$) | **33,969,968** | 661,386,777 | 467,282,853 | 194,103,710 | 3,223 |
| iPDB | A*+BFHS (4) | 37,090,062 | 547,745,706 | 397,125,094 | 150,620,398 | 2,694 |
| *logistics00* | A* | 82,161,805 | 167,974,727 | 163,970,672 | 4,004,055 | **663** |
| 15-1 | BFIDA* | **13,638,319** | 2,847,571,079 | 163,970,672 | 2,660,698,165 | 19,062 |
| $C^*$=67 | A*+BFHS ($\infty$) | 18,827,830 | 730,154,067 | 722,390,335 | 7,763,336 | 4,897 |
| iPDB | A*+BFHS (4) | 18,827,830 | 251,960,077 | 198,537,096 | 53,422,585 | 1,627 |
| *pipesworld-* | A* | 123,553,926 | 284,884,903 | 284,880,335 | 4,568 | **727** |
| *notankage* 19 | BFIDA* | 86,818,434 | 1,227,115,669 | 634,454,295 | 576,633,809 | 4,140 |
| $C^*$=24 | A*+BFHS ($\infty$) | **42,192,503** | 619,095,459 | 619,013,855 | 81,147 | 2,072 |
| iPDB | A*+BFHS (4) | 44,706,153 | 574,957,328 | 570,451,612 | 4,505,259 | 1,942 |
| *parking14* | A* | 351,976,816 | 828,472,606 | 828,472,562 | 44 | **971** |
| 16_9-01 | BFIDA* | 183,832,715 | 4,846,132,188 | 1,023,897,982 | 3,821,980,237 | 6,236 |
| $C^*$=24 | A*+BFHS ($\infty$) | **30,675,587** | 1,191,570,432 | 1,191,514,776 | 55,283 | 1,468 |
| iPDB | A*+BFHS (4) | 51,147,740 | 1,013,776,888 | 1,011,227,268 | 2,549,247 | 1,290 |
| *visitall11* | A* | 407,182,291 | 795,670,561 | 795,669,929 | 632 | **1,045** |
| 08-half | BFIDA* | 172,474,497 | 3,159,596,842 | 1,332,828,069 | 1,824,866,109 | 4,220 |
| $C^*$=43 | A*+BFHS ($\infty$) | **34,406,966** | 1,639,641,152 | 1,639,585,228 | 55,798 | 2,233 |
| iPDB | A*+BFHS (4) | 64,671,078 | 1,346,690,454 | 1,312,333,974 | 34,356,354 | 1,902 |

Table 1: Instances sorted by A* running times. An underline means more than 8 GB of memory was needed. Smallest memory and shortest times are in boldface.

| Instance | Algorithm | Peak stored | Total nodes | Prev. iterations | Last iteration | Time (s) |
|---|---|---|---|---|---|---|
| *tidybot11* | A* | 115,965,857 | 246,756,618 | 246,756,201 | 417 | **1,086** |
| 16 | BFIDA* | 86,095,996 | 1,090,011,154 | 652,777,121 | 431,816,881 | 5,512 |
| $C^*$=40 | A*+BFHS ($\infty$) | **41,342,908** | 583,309,116 | 570,082,820 | 13,225,950 | 2,923 |
| iPDB | A*+BFHS (4) | 57,026,598 | 598,365,499 | 519,723,294 | 78,641,859 | 3,080 |
| *snake18* | A* | 94,699,640 | 129,288,606 | 129,273,608 | 14,998 | **1,131** |
| 08 | BFIDA* | 44,231,998 | 1,852,488,086 | 1,517,078,892 | 325,204,785 | 14,877 |
| $C^*$=58 | A*+BFHS ($\infty$) | **44,081,853** | 391,010,354 | 390,681,641 | 328,706 | 3,445 |
| iPDB | A*+BFHS (4) | 51,166,308 | 356,988,514 | 348,015,242 | 8,973,265 | 3,192 |
| *hiking14* | A* | 287,192,625 | 3,299,939,168 | 3,299,937,850 | 1,318 | **1,297** |
| 2-2-8 | BFIDA* | **42,570,885** | 11,376,337,161 | 5,757,334,602 | 5,582,502,874 | 10,847 |
| $C^*$=42 | A*+BFHS ($\infty$) | 44,454,322 | 16,233,911,987 | 12,346,881,620 | 3,886,689,991 | 14,897 |
| M&S | A*+BFHS (4) | 53,148,260 | 9,850,751,126 | 6,310,295,933 | 3,540,114,817 | 9,696 |
| *pipesworld-* | A* | 292,998,092 | 907,283,307 | 907,283,301 | 6 | **1,364** |
| *tankage* 14 | BFIDA* | 158,262,429 | 5,354,342,623 | 3,680,871,467 | 1,661,344,123 | 10,609 |
| $C^*$=38 | A*+BFHS ($\infty$) | **84,077,693** | 5,768,933,724 | 5,763,927,002 | 5,002,176 | 11,622 |
| iPDB | A*+BFHS (4) | 103,288,306 | 3,300,541,977 | 3,220,772,288 | 79,765,143 | 6,896 |
| *blocks* | A* | 555,864,249 | 1,185,065,570 | 205,172,261 | 979,893,309 | **1,540** |
| 13-1 | BFIDA* | 99,782,317 | 1,742,819,669 | 463,603,038 | 1,224,383,750 | 2,142 |
| $C^*$=44 | A*+BFHS ($\infty$) | **54,601,577** | 2,261,321,708 | 425,991,501 | 1,827,341,160 | 2,817 |
| iPDB | A*+BFHS (4) | 79,572,108 | 1,817,197,763 | 401,559,990 | 1,407,648,726 | 2,317 |
| *parking14* | A* | 606,117,759 | 1,430,911,954 | 1,430,746,610 | 165,344 | **1,714** |
| 16_9-03 | BFIDA* | 291,822,896 | 8,077,642,530 | 1,796,305,162 | 6,280,923,558 | 10,059 |
| $C^*$=24 | A*+BFHS ($\infty$) | **48,304,204** | 2,519,414,336 | 2,328,368,930 | 191,043,484 | 3,124 |
| iPDB | A*+BFHS (4) | 63,455,874 | 2,151,415,198 | 1,992,188,756 | 159,224,520 | 2,679 |
| *tidybot11* | A* | 175,574,760 | 372,772,055 | 372,771,560 | 495 | **1,730** |
| 18 | BFIDA* | 114,747,861 | 1,718,896,347 | 1,093,273,564 | 613,928,542 | 8,810 |
| $C^*$=44 | A*+BFHS ($\infty$) | **40,540,308** | 1,045,166,148 | 1,028,635,660 | 16,529,544 | 5,410 |
| iPDB | A*+BFHS (4) | 65,784,369 | 1,204,942,101 | 931,501,196 | 273,439,961 | 6,365 |
| *blocks* | A* | 704,938,102 | 1,568,547,017 | 342,339,737 | 1,226,207,280 | **1,990** |
| 13-0 | BFIDA* | 137,821,868 | 2,421,546,636 | 775,076,076 | 1,628,338,675 | 2,977 |
| $C^*$=42 | A*+BFHS ($\infty$) | **81,918,224** | 3,498,922,607 | 774,231,514 | 2,710,189,950 | 4,483 |
| iPDB | A*+BFHS (4) | 126,629,640 | 2,615,897,101 | 698,028,054 | 1,903,367,904 | 3,378 |
| *hiking14* | A* | 368,433,117 | 6,711,042,999 | 6,710,971,209 | 71,790 | **2,480** |
| 2-3-6 | BFIDA* | **124,686,777** | 38,476,138,468 | 29,175,130,389 | 8,123,329,545 | 42,379 |
| $C^*$=28 | A*+BFHS ($\infty$) | 146,623,619 | 107,138,328,055 | 106,429,883,507 | 682,558,443 | 120,494 |
| M&S | A*+BFHS (4) | 148,357,537 | 68,496,320,172 | 65,779,382,852 | 2,691,051,215 | 76,603 |
| *pipesworld-* | A* | 442,232,520 | 1,028,882,844 | 1,028,880,896 | 1,948 | **2,693** |
| *notankage* 20 | BFIDA* | 301,349,348 | 4,454,789,871 | 2,384,958,671 | 2,032,377,777 | 15,245 |
| $C^*$=28 | A*+BFHS ($\infty$) | **133,708,317** | 3,325,668,014 | 3,267,529,384 | 58,132,775 | 11,499 |
| iPDB | A*+BFHS (4) | 148,029,967 | 2,988,248,448 | 2,728,140,813 | 260,097,006 | 10,629 |
| *snake18* | A* | 265,033,991 | 367,639,596 | 365,927,487 | 1,712,109 | **3,967** |
| 17 | BFIDA* | 60,041,363 | 2,162,411,969 | 1,464,995,207 | 639,565,966 | 20,418 |
| $C^*$=62 | A*+BFHS ($\infty$) | **56,839,243** | 877,934,374 | 871,327,013 | 6,607,339 | 8,785 |
| iPDB | A*+BFHS (4) | 73,365,792 | 855,342,127 | 776,892,002 | 78,450,103 | 8,916 |
| *satellite* | A* | 107,395,076 | 463,747,690 | 463,744,251 | 3,439 | **11,834** |
| 08 | BFIDA* | 20,846,202 | 3,656,980,017 | 520,525,131 | 3,125,446,334 | 398,884 |
| $C^*$=26 | A*+BFHS ($\infty$) | **18,870,254** | 552,221,751 | 551,990,933 | 230,549 | 54,551 |
| LM-cut | A*+BFHS (4) | 19,763,323 | 546,211,783 | 479,810,475 | 66,401,039 | 56,296 |

Table 2: Instances sorted by A* running times. An underline means more than 8 GB of memory was needed. Smallest memory and shortest times are in boldface.

the last iteration of A*+BFHS is usually only a small portion of the total nodes generated, so the large difference in the last iteration is not very important. A*+BFHS (4) stored a larger maximum number of nodes than A*+BFHS ($\infty$) on almost all instances. However, the difference was usually small and never more than a factor of two. For the running time, the difference was usually less than 50%. Compared to A*+BFHS ($\infty$), A*+BFHS (4) was faster by a factor of 3 on logistics00 15-1, 2.5 on rovers 09 and 11, 4.6 on termes18 05, 3.9 on tpp 11, and never more than 30% slower.

In general, making each call to BFHS on frontier nodes at multiple depths increases both the memory usage and the number of nodes generated in the last iteration, but reduces the number of duplicate nodes and hence is often faster. Con-

| Instance | Algorithm | Peak stored | Total nodes | Prev. iterations | Last iteration | Time (s) |
|---|---|---|---|---|---|---|
| *blocks* | A* (unfinished) | >814,951,324 | >1,562,632,802 | 256,247,910 | >1,306,384,892 | >2,284 |
| 15-0 | BFIDA* | 113,471,990 | 2,408,362,561 | 579,842,889 | 1,827,125,272 | **3,058** |
| $C^*$=40 | A*+BFHS ($\infty$) | **68,070,197** | 3,861,465,924 | 550,007,126 | 3,291,490,500 | 4,889 |
| iPDB | A*+BFHS (4) | 106,482,059 | 2,656,641,036 | 492,390,560 | 2,144,282,178 | 3,514 |
| *storage* | A* (unfinished) | >799,907,374 | >1,741,590,894 | >1,741,590,894 | | >2,358 |
| 17 | BFIDA* | 397,798,456 | 13,297,651,168 | 4,430,334,119 | 8,825,291,425 | 19,086 |
| $C^*$=26 | A*+BFHS ($\infty$) | **118,138,352** | 13,403,671,261 | 13,364,290,422 | 39,380,047 | 18,914 |
| iPDB | A*+BFHS (4) | 133,800,503 | 7,895,157,984 | 6,819,827,727 | 1,075,329,465 | **11,354** |
| *driverlog* | A* (unfinished) | >786,467,847 | >2,028,764,217 | >2,028,764,217 | | >1,853 |
| 15 | BFIDA* | 453,643,579 | 24,705,660,389 | 6,388,627,692 | 18,280,039,412 | 24,297 |
| $C^*$=32 | A*+BFHS ($\infty$) | **88,449,751** | 16,928,608,100 | 16,913,831,869 | 14,773,242 | 15,311 |
| iPDB | A*+BFHS (4) | 123,602,679 | 9,160,294,407 | 8,974,814,158 | 185,477,260 | **8,447** |
| *rovers* | A* (unfinished) | >801,124,989 | >4,427,878,559 | >4,427,878,559 | | >2,776 |
| 09 | BFIDA* | 235,386,020 | 20,666,689,222 | 7,239,737,785 | 13,401,874,237 | 25,336 |
| $C^*$=31 | A*+BFHS ($\infty$) | **96,100,365** | 34,236,064,765 | 34,235,937,332 | 123,597 | 42,290 |
| iPDB | A*+BFHS (4) | 99,498,513 | 12,845,107,625 | 12,752,327,728 | 92,776,061 | **16,770** |
| *rovers* | A* (unfinished) | >766,016,316 | >3,690,650,688 | >3,690,650,688 | | >2,378 |
| 11 | BFIDA* | 274,612,697 | 18,975,576,425 | 6,574,504,656 | 12,391,406,745 | 26,022 |
| $C^*$=30 | A*+BFHS ($\infty$) | **112,783,085** | 32,143,105,562 | 32,139,546,138 | 3,549,575 | 43,538 |
| iPDB | A*+BFHS (4) | 113,594,902 | 12,342,784,453 | 11,789,007,437 | 553,767,167 | **16,661** |
| *parking14* | A* (unfinished) | >770,874,998 | >1,681,926,228 | >1,681,926,228 | | >2,306 |
| 16_9-04 | BFIDA* | 1,045,614,854 | 27,924,183,007 | 6,292,017,194 | 21,628,727,845 | 37,701 |
| $C^*$=26 | A*+BFHS ($\infty$) | **156,758,802** | 9,778,837,190 | 9,777,264,498 | 1,570,687 | 12,304 |
| iPDB | A*+BFHS (4) | 181,535,647 | 7,588,132,706 | 7,586,728,152 | 1,402,549 | **9,813** |

Table 3: Instances where A* terminated without solving the problem (marked by >) so are sorted by BFIDA* running times. An underline means more than 8 GB of memory was needed. Smallest memory and shortest times are in boldface.

sidering the memory and time trade-off, given a new problem, we recommend making each call to BFHS on frontier nodes at multiple depths. So far, we have only tested limiting BFHS to four calls in each iteration. Determining the optimal number of calls to BFHS is a subject for future work.

## Heuristic Functions and Running Times

For each node generated, A* first does duplicate checking then looks up its heuristic value if needed. Thus for each state, A* only computes its heuristic value once, no matter how many times this state is generated. However, the situation is different in BFHS. Even in a single call to BFHS, a state's heuristic value may be calculated multiple times. For example, if a state's $f$-value is greater than the cost bound of BFHS, then this state is never stored in this call to BFHS and its heuristic value has to be computed every time it is generated. In addition, A* has only one hash map but our BFHS implementation has one hash map for each layer of nodes. Consequently, for each node generated, A* does only one hash map lookup while BFHS may have multiple lookups.

Due to the above differences, the number of nodes generated per second of BFIDA* and A*+BFHS was smaller than that of A*. For the iPDB and M&S heuristics, this difference was usually less than a factor of two. For the LM-cut heuristic, A* was faster by a factor of four in terms of nodes generated per second on the satellite domain. This is because computing a node's LM-cut heuristic is much more expensive than iPDB and M&S heuristics. This contrast shows that the choice of heuristic function also plays an important role in comparing the running times of different algorithms.

## Future Work

Future work includes the following. First, test A*+BFHS on more unit-cost domains. Second, investigate what is the best memory threshold for the A* phase. Third, determine the optimal number of calls to BFHS in each iteration. Fourth, find other ways to partition the frontier nodes besides the current depth-based approach. If a set of frontier nodes is too large, we may split it into multiple smaller sets and make one call to BFHS on each such smaller set. This approach may reduce the maximum number of stored nodes but may generate more duplicate nodes. In addition, when we make each call to BFHS on frontier nodes at multiple depths, we may consider the number of frontier nodes at each depth so each call to BFHS is on a different number of depths instead of a fixed number. Fifth, find out how to apply A*+BFHS to domains with non-unit operator costs. For such domains, BFHS's BFS can be replaced by uniform-cost search or Dijkstra's algorithm (Dijkstra 1959). In this case, we can store nodes with multiple costs in each layer (Zhou and Hansen 2006). Sixth, use external memory such as magnetic disk or flash memory in A*+BFHS to solve very hard problems. For example, instead of allocating 1/10 of RAM for the A* phase, we can first run A* until RAM is almost full, then store both Open and Closed nodes in external memory and remove them from RAM. Then in the BFHS phase, we load back the set of frontier nodes for each call to BFHS from external memory. This A*+BFHS version would never perform worse than A*, since it is identical to A* until memory is exhausted, at which point the BHFS phase would begin.

## Conclusions

We introduce a hybrid heuristic search algorithm A*+BFHS for solving hard problems that cannot be solved by A* due to memory limitations, or IDA* due to the existence of many short cycles. A*+BFHS first runs A* until a user-specified storage threshold is reached, then runs multiple iterations of BFHS on the frontier nodes, which are the Open nodes at the end of the A* phase. Each iteration has a unique cost bound and contains multiple calls to BFHS. Each call to BFHS within the same iteration has the same cost bound but a different set of frontier nodes to start with. Within an iteration, frontier nodes are sorted deepest-first so that A*+BFHS can terminate early in its last iteration.

On the around 500 easy problems solved, A*+BFHS behaves the same as A*, and is always faster than BFIDA*. On the 32 hard instances presented, A*+BFHS is slower than A* but uses significantly less memory. A*+BFHS is faster than BFIDA* on 27 of those 32 instances and at least twice as fast on 16 of those. Furthermore, A*+BFHS requires less memory than BFIDA* on 25 of those 32 instances and saves more than half the memory on 14 of those. Another contribution of this paper is a comprehensive testing of BFIDA* on many planning domains, which is lacking in the literature.

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
