# OpenReview forum: "A*+BFHS: A Hybrid Heuristic Search Algorithm"
_icaps-conference.org/ICAPS/2021/Workshop/HSDIP — HSDIP 2021_

### Official Review · AnonReviewer1 · 2021-05-25

**Confidence:** 3
**Overall Score:** Accept

**Review:**

This paper presents an algorithm that performs an A* search until a user-specified memory limit is reached, and then performs (multiple) breadth-first heuristic searches (BFHS). The main advantage of this hybrid search is that it behaves like A* on smaller instances where the exponential memory consumption of A* is not a problem, but at the same time can solve larger, more memory-intensive tasks by using BFHS. Advantages and disadvantages of this new hybrid search algorithm are described and discussed in comparison to related algorithms.  An empirical evaluation on "hard" (memory-intensive) unit-cost planning tasks shows that A*+BFHS performs favorably over A* and Breadth-First Iterative-Deeping-A*.

The topic of the paper fits the workshop, as one of the characteristic topics of the HSDIP workshop is the study of "novel search techniques for domain-independent planning". The newly presented algorithm is well motivated and shows good empirical performance.

Concerning the optimality and completeness of A*+BFHS, I suppose that it follows directly from the construction if an admissible heuristic is used. However, I missed such a statement or proof in the paper. At least I think it would be important to mention this somewhere in the paper so that a reader can directly see that these properties hold. I assume that an admissible but inconsistent heuristic is sufficient if reopening is performed by all the heuristic searches involved? Otherwise, the landmark cut heuristic used in some of the experiments would be problematic.

Speaking of optimality with an admissible heuristic: In some places it was difficult to understand why the generated solutions are optimal with an admissible but inconsistent heuristic. In particular, in the section on solution reconstruction. It states, "We use A* to compute the path from the start node to the middle node using the same heuristic function for the original problem, which measures the distance to the goal node, not the middle node." At first glance, it is not entirely clear to me under what termination criteria this search reconstructs the optimal path to the middle node, especially when an inconsistent heuristic is used.

With regards to the empirical evaluation, is there a specific reason why some heuristics were chosen for certain domains?
Even though I see some value in the detailed table, the large amount of numbers is somewhat difficult to analyze at once. I would suggest visualizing some of the data as diagrams or plots (personal preference).

Overall, I think the presented hybrid heuristic algorithm (A*+BFHS) is a valuable contribution to the workshop, so I recommend accepting the paper.

---

> ### Author Response · Authors · 2021-06-08
> **Review 1 response**
>
> Thank you very much for your hard work and careful review.
>
> > Concerning the optimality and completeness of A*+BFHS, I suppose that it follows directly from the construction if an admissible heuristic is used. However, I missed such a statement or proof in the paper. At least I think it would be important to mention this somewhere in the paper so that a reader can directly see that these properties hold. I assume that an admissible but inconsistent heuristic is sufficient if reopening is performed by all the heuristic searches involved? Otherwise, the landmark cut heuristic used in some of the experiments would be problematic.
>
> Yes, the algorithm is admissible and complete when using an admissible heuristic. These properties do not require the heuristic functions to be consistent, so an admissible but inconsistent heuristic function is sufficient. We will add a statement in our final version about these properties.
>
> > Speaking of optimality with an admissible heuristic: In some places it was difficult to understand why the generated solutions are optimal with an admissible but inconsistent heuristic. In particular, in the section on solution reconstruction. It states, "We use A* to compute the path from the start node to the middle node using the same heuristic function for the original problem, which measures the distance to the goal node, not the middle node." At first glance, it is not entirely clear to me under what termination criteria this search reconstructs the optimal path to the middle node, especially when an inconsistent heuristic is used.
>
> For solution reconstruction, the specific sentence in question is for BFIDA*. As we have mentioned in the response to review 2, the original BFIDA* solution reconstruction requires heuristic functions that can return the heuristic values between any two nodes. This is not possible for heuristic functions like iPDB so we used our own way to reconstruct the solution for BFIDA*. In the following paragraph, we will explain how we constructed the optimal path between the start node to the middle node in our BFIDA* implementation. Bear with us this is a little bit complicated.
>
> In our BFIDA* implementation, when we are reconstructing the solution path between the start node and the middle node, we already know the optimal depth of the middle node (denoted as Mg), so we only need to reconstruct the path. The simplest and most straightforward way is just to run a pure breadth-first search from the start node to the middle node. However, this approach may generate too many nodes. Therefore, we run our modified A* to select nodes to expand, which reduces the number of generated nodes. We run our modified A* from the start node. We use the heuristic function (like iPDB) that returns the heuristic values to the original goal node, which is the only heuristic function we computed before BFIDA* starts. We will expand an Open node n that satisfies two criteria: 1) f(n) <= the optimal solution cost of the original problem (C*), and 2) g(n) < Mg. The optimal path from the start node to the middle node is part of the optimal path to the original goal node. Therefore, every node on the optimal path from the start node to the middle node satisfies the just mentioned two criteria (of course, the middle node itself has g(n) = Mg). Conversely, for an Open node n that has f(n) > C* or g(n) >= Mg, it is guaranteed that n is not on the optimal path from the start node to the middle node, so we can safely prune n. For an Open node n such that g(n) < Mg but f(n) > C*, it is guaranteed that n is not on the optimal path from the start node to the original goal node, hence not on the optimal path from the start node to the middle node, therefore we can prune n. Our modified A* keeps expanding nodes until it generates the middle node at depth Mg. An admissible but inconsistent heuristic is sufficient here. The worst case is that we expand all nodes above Mg, in which case we expand the same set of nodes as pure breadth-first search.
>
> > With regards to the empirical evaluation, is there a specific reason why some heuristics were chosen for certain domains?
>
> For each domain, we tested all three heuristics and we used the best one in our experiments.
>
> > I would suggest visualizing some of the data as diagrams or plots (personal preference).
>
> Thank you for recommending the plots. We will add plots in our camera-ready version.

---

### Official Review · AnonReviewer2 · 2021-05-26

**Confidence:** 4
**Overall Score:** Accept

**Review:**

The paper presents a new algorithm for solving instances where A* runs out of
memory. The algorithm first performs A* for 1/10 of the available memory, then
performs iterative breadth-first heuristic search (BFHS), which utilizes
heuristics to prune states whose f-value is over the given bound. When A* is cut
off, the lowest f-value in the frontier is used as initial bound, and all
frontier states are split into different sets depending on their depth. An
iteration of BFHS then performs several BFHS searches, one for each state set
(which serves as initial open list).
The experimental evaluation shows that this approach strikes a good balance
between lowering memory requirements and still keeping enough nodes for useful
duplicate checking.

Overall the topic of the paper is a good fit for HSDIP, it is well written,
novel (as far as I can tell), clear, and while the set of benchmarks is rather
small it is sufficiently large for a workshop paper and the discussion of the
results convincingly shows the advantages of the approach. A clear accept in my
opinion.

One minor thing that is missing is the statement that the algorithm is correct,
complete and optimal. While I do believe that it has these properties I think
that it is important to state it and give at least a very high-level reason why
this is the case.

I also have two remarks regarding readability. I don't expect you to address
them for a final version for this workshop, but maybe you find them helpful if
you pursue to publish the paper in a conference or journal.
1) Consider to have high-level pseudocode for A*+BFHS. You describe it in
   sufficient detail in the text, but I often jumped back to the algorithm
   description at a later point because I wanted to remind myself how certain
   parts worked and pseudocode would be more helpful for these cases.
2) While the tables are very detailed it is hard to grasp anything from them
   "at a glance" and I didn't check all entries in detail. I believe that in
   many cases a plot (like for example a scatterplot comparing total # nodes or
   max # of concurrently stored nodes for different configurations) would
   convey the information in a much more accessible way.

I also have the following questions to the authors and would appreciate it if
they would briefly elaborate these points in a final version of the paper:
1) You mention several algorithms that store some form of middle layer (for
   solution reconstruction). How do they decide what the "middle" layer is? Half
   of the heuristic value of the initial state?
2) What is the max-tree first and longest-path-first policy (mentioned when
   comparing A*+BFHS to FPS)? What effect does the different policies have on for
   example the search space of the last iteration?


Minor remarks:
 - Introduction: At this point the reader might not know what BFIDA* is; either
 explain it shortly or at least mention that is is another algorithm trying to
 tackle the same problem. Also, the sentence is almost identical to the last
 sentence of the abstract.
 - page 3, second paragraph: "we can decrease ... by making each call to BFHS
 on frontier nodes at adjacent depths" -> by combining sets of states with
 adjacent depths? The original sentence seems weird and unclear.
 - Solution reconstruction: "We use A* to compute the path from the start node
 to the middle node..." -> I was at first confused whether BFIDA* is your
 contribution or not, because it sounds like you came up with the things you
 describe. Are you describing the original BFIDA* solution reconstruction or
 did you alter it?
 - Experimental Results and Analysis, first paragraph: "complicate" -> complicated
 - Calling BFHS on Nodes at Multiple Depths: "and never 30% slower" -> never
 more than 30% slower?
 - References: page numbers missing for Asai and Fukunaga 2016; Helmert and
 Domshlak 2009; Sievers 2018; Sievers, Ortlieb and Helmert 2012; Sievers,
 Wehrle and Helmert 2014; Sievers, Wehrle and Helmert 2016.
 - References: You often write "volume x" for Conferences instead of "xth
 Conference on ..." which I find unusual.
 - References, Sievers 2018: missing venue
 - References, Martines et al 2108: ipc -> IPC

---

> ### Author Response · Authors · 2021-06-08
> **Review 2 response**
>
> Thank you very much for your hard work and careful review.
>
> > One minor thing that is missing is the statement that the algorithm is correct, complete and optimal. While I do believe that it has these properties I think that it is important to state it and give at least a very high-level reason why this is the case.
>
> Yes, the algorithm is admissible and complete when using an admissible heuristic. Both A* and pure BFHS have those properties. Our algorithm feeds all Open nodes at the end of the A* phase to calls to BFHS. Therefore, when an optimal solution exists, one node on this optimal path will serve as one of the start nodes for one of the calls to BFHS. The existence of such a node is guaranteed by A*'s completeness and admissibility. Then when the cost bound equals the optimal solution cost C*, the optimal solution will be found, guaranteed by BFHS's completeness and admissibility. We do have formal proofs for our algorithm's completeness and admissibility, but they are too long to be included in the paper. We will add a statement in our final version about these properties.
>
> > Consider to have high-level pseudocode for A*+BFHS.
>
> We will add pseudocode if the space permits.
>
> > I believe that in many cases a plot (like for example a scatterplot comparing total # nodes or max # of concurrently stored nodes for different configurations) would convey the information in a much more accessible way.
>
> Thank you for recommending the plots. We will add plots in our camera-ready version.
>
> > You mention several algorithms that store some form of middle layer (for solution reconstruction). How do they decide what the "middle" layer is? Half of the heuristic value of the initial state?
>
> For DCFA*, we can save nodes at depth h(start)/2 as the middle layer. For BFIDA*, we can save nodes at depth cost_bound/4 as the middle layer. These choices are just recommendations, not requirements.
>
> > What is the max-tree first and longest-path-first policy (mentioned when comparing A*+BFHS to FPS)? What effect does the different policies have on for example the search space of the last iteration?
>
> For FPS, the max-tree-first policy means that at the beginning of each iteration, we first expand the frontier node that generated the most nodes below it in the previous iteration. The longest-path-first policy means at the beginning of each iteration, we first expand the frontier node that generated the deepest leaf node (longest path) below it in the previous iteration. These different policies affect the number of nodes generated in the last iteration. A good policy means we may find the goal early in the last iteration.
>
> > At this point the reader might not know what BFIDA* is; either explain it shortly or at least mention that is is another algorithm trying to tackle the same problem. Also, the sentence is almost identical to the last sentence of the abstract.
>
> We will find a better way to describe BFIDA* in the Introduction section.
>
> > page 3, second paragraph: "we can decrease ... by making each call to BFHS on frontier nodes at adjacent depths" -> by combining sets of states with adjacent depths? The original sentence seems weird and unclear.
>
> Yes. We will revise our words in the final version.
>
> > Are you describing the original BFIDA* solution reconstruction or did you alter it?
>
> The original BFIDA* solution reconstruction requires heuristic functions that can return the heuristic values between any two nodes. This is not possible for heuristic functions like the incremental PDB (iPDB). PDB heuristic is essentially a lookup table that stores heuristic values to the goal node, generated by running a complete search from the abstracted goal node in the abstract search space. As a result, PDB heuristics cannot return heuristic values between any two nodes. So we used our own way to reconstruct the solution for BFIDA*.
>
> > Calling BFHS on Nodes at Multiple Depths: "and never 30% slower" -> never more than 30% slower?
>
> Yes.
>
> Thanks for pointing out all the problems in the References. We used references tex source code from Google Scholar. We will fix them in our final version.

---

> > ### Comment · AnonReviewer2 · 2021-06-08
> > **Thank you for your response**
> >
> > Thank you for your detailed answers and explanations, they were very helpful. Even though you mention the problems with heuristics that only work with goal states I wasn't aware that the original BFIDA* did not consider this issue; so it might be helpful to add this in the final version of the paper.

---

### Decision · Program_Chairs · 2021-06-10

**Decision:**

Accept

**Comment:**

Congratulations, both reviewers agree that the paper is a clear accept. We encourage you to follow up on the suggestions made in the reviews, particularly on the plots if space permits.